# Patient-derived oral mucosa organoids as an *in vitro* model for methotrexate induced toxicity in pediatric acute lymphoblastic leukemia

**E. Driehuis**[1][☯], **N. Oosterom**[2][☯], **S. G. Heil**[3], **I. B. Muller**[4], **M. Lin**[4], **S. Kolders**[1], **G. Jansen**[5], **R. de Jonge**[4], **R. Pieters**[2], **H. Clevers**[1,2], **M. M. van den Heuvel-Eibrink**[2]*

1 Oncode Institute, Hubrecht Institute, Royal Netherlands Academy of Arts and Sciences (KNAW) and University Medical Center Utrecht, Utrecht, The Netherlands, 2 Princess Máxima Center for Pediatric Oncology, Utrecht, The Netherlands, 3 Department of Clinical Chemistry, Erasmus MC, University Medical Center Rotterdam, Rotterdam, The Netherlands, 4 Department of Clinical Chemistry, Amsterdam UMC, Amsterdam, The Netherlands, 5 Amsterdam Rheumatology and Immunology Center, Department of Rheumatology, Amsterdam UMC, Amsterdam, The Netherlands

☯ These authors contributed equally to this work.
* m.m.vandenheuvel-eibrink@prinsesmaximacentrum.nl

**Data Availability Statement:** All data are described within the current manuscript as it is.

## Abstract

We have recently established a protocol to grow wildtype human oral mucosa organoids. These three-dimensional structures can be maintained in culture long-term, do not require immortalization, and recapitulate the multilayered composition of the epithelial lining of the oral mucosa. Here, we validate the use of this model to study the effect of Leucovorin (LV) on Methotrexate (MTX)-induced toxicity. MTX is a chemotherapeutic agent used in the treatment of pediatric acute lymphoblastic leukemia. Although effective, the use of MTX often results in severe side-effects, including oral mucositis, which is characterized by epithelial cell death. Here, we show that organoids are sensitive to MTX, and that the addition of LV reduces MTX toxicity, in both a concentration- and timing-dependent manner. Additionally, we show that a 24 hour 'pretreatment' with LV reduces MTX-induced cell death, suggesting that such a pretreatment could decrease mucositis in patients. Taken together, we provide the first *in vitro* model to study the effect of MTX on wildtype oral mucosa cells. Our findings underscore the relevance of the clinically applied LV regimen and highlight the potential of this model to further optimize modifications in dosing and timing of Leucovorin on oral mucosa cells.

## Introduction

High-dose methotrexate (HD-MTX) is an important antifolate chemotherapeutic agent used in pediatric acute lymphoblastic leukemia (ALL) therapy. Currently, five-year survival rates of pediatric ALL have reached 90% in developed countries [1–4]. However, patients often suffer from MTX toxicities such as hepatotoxicity, nephrotoxicity, hematological malignancies and

**Funding:** Funded by the Oncode Institute (partly financed by the Dutch Cancer Society), by the gravitation program CancerGenomiCs.nl from the Netherlands Organization for Scientific Research (NWO) and by a Stand Up to Cancer International Translational Cancer Research Grant, a program of the Entertainment Industry Foundation administered by the AACR (SU2C-AACR-DT1213) and a ZonMw grant (116.006.103).

**Competing interests:** H.C. is an inventor on several patents related to organoid technology. E.D. is inventor on a patent related to the culturing of organoids derived of normal and tumor tissue of the head and neck area. The relevant patent numbers are: WO2009/022907, WO2010/090513, WO2012/168930. This does not alter our adherence to PLOS ONE policies on sharing data and materials.

intestinal and oral mucositis [5]. Despite administration of folinic acid (Leucovorin—LV) after HD-MTX infusion, 20% of patients develop severe HD-MTX-induced oral mucositis leading to chemotherapy delays and an impaired quality of life [5–7]. The development of oral mucositis is a complex process, of which therapy-induced epithelial cell death is one of the main features [8, 9].

After entering the cell (via reduced folate carrier 1 (RFC1), proton-coupled folate transporter (PCFT) or membrane folate receptors (MFR)), MTX is polyglutamated (PG) by folylpolyglutamate synthetase (FPGS) [10, 11]. This polyglutamation is essential, as it increases intracellular MTX retention and augments its pharmacological activity [12]. MTX-PG inhibits DNA and RNA synthesis via inhibition of dihydrofolate reductase (DHFR), thymidylate synthase (TS) and aminoimidazole carboxamide ribonucleotide transformylase (AICART-Fase), resulting in depletion of intracellular reduced folate levels (see S1 Fig for an overview of MTX transport, metabolism and action) [13]. Ultimately, this results in apoptosis in leukemia cells. However, also healthy cells with a high cell turnover, including the bone marrow and the epithelial lining of the gastrointestinal tract and oral mucosa, are affected by MTX therapy.

LV is a reduced folate that is able to restore purine/pyrimidine biosynthesis after HD-MTX therapy [14, 15]. Clinical guidelines advise to administer LV after HD-MTX to reduce toxicity [16–22]. While it is nowadays generally accepted that LV rescue therapy decreases MTX-induced toxicities such as oral mucositis after MTX administration, the optimal LV dosing- and timing- regimen to reduce oral mucositis rates remains unknown and varies throughout the world.

Although LV rescue therapy was already introduced in the 1960s, preclinical studies have only retrospectively provided a biochemical rationale for the efficacy of this therapy. In mice, a decrease in MTX-induced damage to the jejunal- and oral mucosa was observed after LV administration [23–26]. Importantly, when administered with a time-interval of 12 to 24 hours after MTX, LV did not compromise the anti-leukemic activity of MTX [23–26]. In line with this, a selective mechanism of action for MTX and LV in tumor cells versus normal healthy cells has been proposed. A higher level of MTX-PG was observed in leukemia- and solid tumor cell lines when compared to normal intestinal and bone marrow precursor cells, both *in vitro* and *in vivo* [16–22]. In contrast, several pediatric ALL studies [27–30] have suggested that Leucovorin rescue therapy decreases toxicity rates, but might be accompanied by an increased risk of relapse in ALL. This phenomenon has been referred to as the folate 'over-rescue' principle, where not only healthy cells, but also tumor cells are rescued. There are no studies that investigate the effect of MTX and LV in healthy human oral mucosa cells and, although valuable, it has been acknowledged that 2D cell lines or mouse models do not always reliably predict the clinical utility of tested therapies [31, 32]. Taken together, there is a need for models that more closely recapitulate the *in vivo* situation and would allow clinicians and researchers to investigate the effects of altering LV administration regimens on MTX-induced toxicity in healthy mucosal cells.

Organoids are 3D structures grown from stem cells, that recapitulate histological and functional characteristics of their tissue of origin [33]. Since the discovery that organoids could be established from adult stem cells of the mouse gut, organoid technology has quickly evolved [34]. Nowadays, organoids can be grown from many different epithelia [34–46]. These 'mini-organs' can be established from both tumor and normal primary patient material with high (60–70%) efficiency. Data supporting the translational potential of this technology is accumulating. For example, *in vitro* therapy response of tumor organoids was shown to predict the responses of corresponding patients [47–52]. When derived from Cystic Fibrosis (CF) patients, organoids were also found to predict patient response *in vitro* [53] and could be used to find effective therapies for CF patients [54].

Recently, we have described an organoid model derived from healthy oral mucosa [38]. The resulting patient-derived structures consist of a functional stratified squamous epithelium that can be maintained and expanded in culture for over six months. Upon passaging, organoids grown from primary oral mucosa tissue can be broken into smaller fragments, which will proliferate and result in the formation of new organoids. As such, organoid technology allows us to multiply human wildtype epithelial cells for a wide variety of applications, including drug screening. Taken that others have shown that organoids are a proper model for body physiology, we set out to test the potential of different dosing- and timing- regimens of LV in search for the most optimal regimen to 'rescue' mucosal toxicity during treatment with HD-MTX in patient-derived oral mucosal organoids.

## Methods

### Establishment and culture of human organoid lines

Tissue for the generation of organoids from adult normal human oral keratinocytes was obtained from tissue biopsies in the oral cavity during ear/nose/throat surgery. Oral mucosa organoids were generated as previously described [38]. Patient material was collected from pathology material in Advanced DMEM/F12 (Life Technologies, cat.no. 12634–034), supplemented with 1x GlutaMAX (adDMEM/F12; Life Technologies, cat.no. 12634–034), Penicillin-streptomycin (Life Technologies, cat.no. 15140–122) and 10 mM HEPES (Life Technologies, cat.no. 15630–056). This medium was called +/+/+ medium. In addition, 100 μg/mL Primocin (Invivogen, cat.no. ant-pm1) was added to the +/+/+ medium for tissue collection. The tissue was cut into small fragments. When macroscopically visible, muscle or fat tissue was removed to enrich for the oral epithelium before digestion. Fragments were incubated at 37˚C in 0.125% Trypsin (Sigma, cat.no. T1426) in +/+/+ medium until digested (+/- 30 minutes, never longer than 60 minutes). After trypsinization and centrifugation, the resulting pellet was resuspended in ice-cold 70% 10 mg·mL$^{-1}$ cold Cultrex growth factor reduced BME type 2 (Trevigen, cat.no. 3533-010-02) in organoid medium. Organoid medium contained 1x B27 supplement (Life Technologies, cat. no. 17504–044), 1.25 mM N-acetyl-L-cysteine (Sigma-Aldrich, cat.no. A9165), 10 mM Nicotinamide (Sigma-Aldrich, cat.no. N0636), 50 ng/mL human EGF (Pepro-Tech, cat.no. AF-100-15), 500 nM A83-01 (PeproTech, cat. no. 100–26), 10 ng/mL human FGF10 (PeproTech, cat.no. 100–26), 5 ng/mL human FGF2 (PeproTech, cat.no. 100-18B), 1 μM Prostaglandin E2 (Tocris Bioscience, cat.no. 2296), 0.3 μM CHIR 99021 (Sigma-Aldrich, cat.no. SML1046), 1 μM Forskolin (Bio-Techne R&D Systems, cat.no. 1099), 4% RSPO and 4% Noggin (produced via r-PEX protein expression platform at U-Protein Express BV). Droplets of approximately 10 μl were plated on the bottom of pre-heated suspension culture plates (Greiner, cat.no. M9312). After plating, plates were inverted and put at 37˚C for 30 minutes to let the BME solidify and to prevent the cells from attaching to the bottom of the plate. Subsequently, prewarmed organoid medium was added to the plate. For the first passage of the newly established organoid line, 10 μM Rho-associated kinase (ROCK) inhibitor Y-27632 (Abmole Bioscience, cat.no. M1817) was added to the medium to aid outgrowth of organoids for the primary tissue. Organoids were split between 7 to 14 days after initial plating. For passaging, organoids were collected from the plate by disrupting the BME droplets with a P1000 and washed in 10 mL +/+/+. The pellet was resuspended in 1 mL of TrypLE Express (Life Technologies, cat.no. 12605–010) and incubated at 37˚C. Digestion was closely monitored and the suspension was pipetted up and down every 5 minutes to aid disruption of the organoids into single cells. Cells were subsequently resuspended in ice-cold 70% BME in organoid medium and plated at suitable ratios (1:5 to 1:20) to allow efficient outgrowth of new organoids. After splitting, 10 μM Y-27632 (Abmole Bioscience, cat. no. M1817) was always added

to aid outgrowth of organoids from single cells. In the experiments, we used organoids from five different donors. Furthermore, we performed drug screens in leukemia cell lines. T cell ALL (MOLT16, HSB2, Jurkat) and B cell ALL (Nalm6, REH) cell lines were obtained from DSMZ-German Collection of Microorganisms and Cell Cultures (DSMZ, Braunschweig, Germany).

## Modification of culture conditions for MTX drug screens

For the purpose of this study, organoids were transferred to medium containing a more physiological concentration of folate rather than media with supra-physiological concentrations of folic acid usually present in regular media. We used RPMI 1640 without folic acid (Thermofisher, cat.no. 27016021) supplemented with the same supplements as in organoid medium supplemented with 5 nM folinic acid (Sigma-Aldrich, cat.no. 47612-250MG; racemic mixture of d- and l-stereoisomer of folinic acid) as sole folate source. This medium was referred to as low folate medium. Medium was changed every 2–3 days and organoids were split once every 1–2 weeks. Organoids were cultured for at least two weeks in this folate-deprived state before starting experiments. All drug screens were performed in this modified, low folate medium.

## Growth curves

Organoids were disrupted into single cells using TrypLE digestion. Cells were counted and 20.000 single cells were plated per well in 24-well plates. Per well, 30 µl BME was plated and subsequently cultured as previously described. To assess growth speed of the culture on different media, each organoid line was cultured in parallel in normal and low folate medium. For each timepoint, organoid material was collected in triplicate (three wells). Material was collected by disrupting the BME drop with a P1000 pipet in a 15 mL falcon tube and 3 mL cold +/+/+ medium was added for washing. After centrifugation, supernatant was removed and pellets were stored at -20˚C until readout. Material was collected at day 0, 3, 5, 7, 10, 12 and 14. For readout, pellets were thawed on ice, and 1 mL of PBS/CellTiter-Glo 3D Reagent (Promega, cat.no. G9681) (1v:1v) was added to the pellet. After a 30 minute incubation on a shaker, 100 µl of the lysate was transferred to a black 96 well plate and luminescence readout was performed to assess cell viability.

## RNA collection

Cells were cultured for a week after splitting before RNA was collected. Two days after splitting, cells were cultured in the presence of 0.5 µM MTX. For collection, the QIAGEN RNA easy kit (Qiagen, cat. no. 74104) was used according to protocol. In short, pellets were collected in 350 µl RLT buffer. Subsequently, 350 µl 70% ethanol was added and mixed by pipetting up and down before transfer to the RNA binding columns provided in the kit. After washing twice with 500 µl of RPE buffer, and once with RW1, columns were centrifuged at maximal speed for 1 minute to assure that they were dry. Elution was performed by the addition of 30 µl RNAase free water. RNA was stored at -80˚C until further use.

## cDNA synthesis

For cDNA synthesis, 10.5 µl RNA was mixed with 1 µl 50 µg/ml 110 diluted Oligo(dT) 15 Primer (Promega, cat.no. C1101) and incubated for 5 minutes at 70˚C. After that, 8.5 GoScript Reverse Transcriptase mastermix (Promega, cat.no. A5003) was added, consisting of RT buffer, MgCl2, dNTPs, RT and RNase inhibitor according to protocol. Samples were incubated

**Table 1.**

| Protein | supplier | Order nr. | Host species | Clone | Lot nr. | Dilution | Antigen retrieval method |
|---------|----------|-----------|--------------|-------|---------|----------|--------------------------|
| TP63 | Abcam | AB735 | Mouse | 4AB | AB735 | 1:800 | Citrate |
| KI67 | Monosan | MONX10293 | Mouse | MM1 | 10293 | 1:2000 | Citrate autoclave |
| KRT13 | Progen | 10523 | Mouse | 1C7 | 10523 | 1:100 | Citrate |

for 5 minutes at 25˚C, 60 minutes at 42˚C and 15 minutes at 72˚C. Samples were stored at -20˚C until use.

## Immunohistochemistry

Tissues or organoids were fixed in 4% paraformaldehyde overnight, dehydrated, and embedded in paraffin. Sections were subjected to H&E as well as IHC staining, using the antibodies shown in Table 1. Stainings were performed at the pathology department of the UMCU (Utrecht, the Netherlands).

## Quantitative PCR for expression of MTX metabolism genes

For quantitative PCR, IQ SYBR green (Bio-Rad, cat.no. 1708880) was used in a 384-well format. Per well, 7.5 µl SYBR Green was used, mixed with 1 µl 10 µM FW primer and 1 µl 10 µM RV primer, 3 µl cDNA mix and 2.5 µl water. For each reaction, it was estimated that 25 ng of cDNA was loaded. For qPCR, samples were incubated for 2 minutes at 95˚C and for 40 cycles at: 15 seconds at 98˚C, 15 seconds at 58˚C and 15 seconds at 72˚C. Results were calculated by using the ΔΔCt method. Expression was calculated relative to expression in tongue tissue (total RNA, human normal tongue tissue, AmsBio, cat.no. R1234267). Melt peak analysis was performed to assure that primer had no aspecific binding. Primers used are described in S4 Table.

## DNA isolation

DNA was isolated using Reliaprep gDNA tissue miniprep system (Promega, cat. no. A2052) according to protocol. DNA concentrations were measured using Nanodrop.

## Whole exome sequencing

Whole exome sequencing of oral mucosa organoids was previously performed [38]. In this study, these data were reported to show the difference in number of detected mutations in both the normal human organoids used in this study compared to the tumor tissue derived from the same patient to establish that we used normal human organoids here.

## LV pretreatment

For LV pretreatment, 24 hours prior to the start of the drugscreen, organoids were exposed to 0.1 µM LV, by replacing the culture media with media containing this concentration of LV. Plates were placed back in the incubator before, 24 hours later, organoids were collected for drug screening as described below.

## MTX drug screens (384-well format)

Two days prior to start of the drug screen, organoids were passaged and disrupted into single cells using TrypLE. Single cells were plated in 70% BME in organoid medium. Two days later, when single cells already reformed into small organoids, the organoids were collected from the

BME by addition of 1 mg/mL dispase II (Sigma-Aldrich, cat.no. D4693) to the medium of the organoids. Organoids were incubated for 30 minutes at 37°C to digest the BME. Subsequently, organoids were washed, filtered using a 70 μm nylon cell strainer (Falcon), counted and resuspended in 5% BME/growth medium (12.500 organoids/mL) prior to plating in 40 μl volume (Multi-drop Combi Reagent Dispenser, Thermo Scientific, cat.no. 5840300) in 384-well plates (Corning, cat.no. 4588). As such, 1000 organoids were plated per well. Drugs were added 1 hour after plating the organoids using the Tecan D300e Digital Dispenser (Tecan).

Methotrexate (Sigma-Aldrich, cat.no. M1000000) was dissolved in DMSO and was used in a concentration range between 5 uM–0.05 uM. Folinic acid (Sigma-Aldrich, cat.no. 47612-250MG) was dissolved in PBS containing 0.3% Tween-20, which was required to dispense the drug using the HP printer, and was used at set concentrations of 0.0125 uM–0.025 uM–0.05 uM–0.1 uM. We based our concentration range of MTX based on median MTX plasma level measured in pediatric ALL patients at T48h (0.38 μM, range 0.1–22 μM). The ratio LV:MTX in clinics is around 1:100 (~50 mg/m$^2$: 5000 mg/m$^2$), which was the rationale to focus on a lower range of LV concentrations (for instance 5 μM MTX versus 0.05 μM LV. All wells were normalized for solvent used. DMSO and percentage PBS/Tween-20 never exceeded 1%. Drug exposure was performed in technical triplicate and biological replicates of at least three for each concentration shown. For a lay-out of the drug screen and morphology of the organoid lines during a drugscreen, see S4 Fig.

For folinic acid (Leucovorin–LV) rescue studies, LV was added at different time points after start of MTX incubation. LV was dispensed using the Tecan Dispenser on top of plates previously started on MTX incubation. No medium change was performed (as organoids are in 5% BME, medium removal is impossible) and LV was dispensed into the medium that contained different concentrations of MTX. LV rescue was performed at 0, 12, 24, 48, 72 and 96 hours.

120 hours after adding the drugs, ATP levels were measured using the CellTiter-Glo 3D Reagent (Promega, cat.no. G9681) according to the manufacturer's instructions and luminescence was measured using a Spark multimode microplate reader (Tecan). Results were normalized to vehicle (no drugs—100% cell viability) and baseline control (Staurosporin 1 μM— 0% cell viability). IC50 values (= half maximal inhibitory concentration; the concentration at which 50% of cells are dead) were calculated for separate experiments. As a quality check for the performed assays, Z-values were calculated for each individual screen. Screens with Z-values below 0.3 were excluded from analysis (S2 Table).

## MTX-polyglutamate analysis by UHPLC-MS/MS

Organoids were plated at a density of 100.000 per 4 mL in a 6-well non-repellent plate (Greiner) in low folate medium. Leukemia cell lines were cultured at a density of $10*10^6$ cells per 20 mL low folate medium supplemented with 10% fetal calf serum (FCS). Organoids or leukemia cell lines were cultured without MTX or with MTX 0.5 μM. After a 24h incubation, cells were collected and washed twice with 15 mL medium (organoids) or PBS (cell lines). After centrifugation, cells were resuspended in 1 mL, counted and then snap frozen. Before counting organoids, they were incubated at 37°C in 0.125% Trypsin (Sigma, cat.no. T1426) until digested to be able to count single cells. Frozen cell or organoid pellets of 2–3 x $10^6$ cells were thawed and resuspended in 50 μl ice-cold PBS (pH 7.4) (B. Braun, Melsungen, Germany) by vortexing. Next, 50 μl $^{13}C_5$$^{15}N$-labeled custom-made stable isotopes of MTX-PG$_{1-7}$ as internal standard (25 nmol/L) and 100 μl perchloric acid (10% v/v, Sigma-Aldrich cat no. 244252) were added and vortexed immediately. [55] Mixtures were incubated on ice for 30 min, followed by centrifugation at 20,160 $x$ $g$ for 15 min at 4°C. Supernatants were quantitatively transferred to an

Eppendorf tube and 34 µl 1M phosphate buffer (pH 11.5) was added during vortexing. 100 µl of the mixture was transferred to 0.22 µm spin columns (Merck Millipore Ltd, cat no UFC30GVNB) attached to an LC vial and centrifuged at 3,200 $x$ $g$ for 10 min at 4˚C. The resulting eluates were subjected to immediate UHPLC-MS/MS analysis of methotrexate monoglutamate to pentaglutamate (MTX-PG$_{1-5}$) by injecting 20 µl samples in an Acquity Ultra Performance LC system (Waters Corporation, Milford, MA, USA) with a Kinetex 1.7 µm EVO C18 (100 Å, 100 x 2.1 mm) LC column maintained at 40˚C. A linear gradient was applied, consisting of solvent A (5% acetonitrile, pH 3.77) and solvent B (55% acetonitrile, pH 3.77). Measurements were performed over total run times of 10 min. The LC system was connected to an AB Sciex 4000 Q Trap tandem quadrupole mass spectrometer (Applied Biosystems, Foster City, CA, USA), which operated in the positive ionization mode with an ion spray voltage of 5.5 kV, collision energy of 40 V, declustering potential of 100 V and collision exit potential of 7 V for all mass transitions. Mass transitions used for the MTX-PG$_{1-5}$ analytes were as follows; MTX-PG$_1$ ($m/z$ = 455.200>308.200), MTX-PG$_2$ ($m/z$ = 584.2>308.2), MTX-PG$_3$ ($m/z$ = 713.3>308.2), MTX-PG$_4$ (m/z = 842.3>308.2) and MTX-PG$_5$ ($m/z$ = 971.4>308.2), respectively. For the $^{13}C_5$$^{15}$N-labeled MTX-PG$_{1-5}$ internal standard, mass transitions used were: $m/z$ = 461.2>308.2, $m/z$ = 590.2>308.2, $m/z$ = 719.3>308.2, $m/z$ = 848.3>308.2 and $m/z$ = 977.4>308.2, respectively. Qualification and integration of the resulting peaks were analyzed with the Analyst software version 1.6.3 (Sciex, Framingham, MA, USA), resulting in the peak area under the curve (AUC). Quantification of MTX-PG$_{1-5}$ concentrations/cell number was performed with the labeled internal standards as described before. [55]

## FPGS activity analysis

FPGS catalytic activity analysis in organoids and leukemic cell lines was performed essentially as described by Muller et al [56]. In short, FPGS protein was isolated from organoids and cell lines by sonication (Sonoplus Mini 20; Bandelin, Berlin, Germany) on ice for 2 · 10 seconds (10 second intervals with 90% amplitude and 30-second intervals between samples) in FPGS extraction buffer (50 mM Tris, 20 mM KCl, 10 mM MgCl2, and 5 mM DTT, pH 7.4, 4oC, in MilliQ (MilliQ Advantage A10; Merck Millipore, Burlington, MA, USA), followed by centrifugation in an Eppendorf centrifuge (12.000 • g, 15 minutes, 4˚C). FPGS-mediated conversion of MTX-PG1 to MTX-PG2 is determined in cell extracts (10–200 µg protein) in a total volume of 250 µL containing final concentrations of 100 mM Tris, 20 mM KCl, 20 mM MgCl2, 10 mM DTT, 10 mM ATP, 250 mM MTX-PG1, and 4 mM 15N-labeled L glutamatic acid (Sigma-Aldrich, cat no. 332143-100MG) at a pH of 8.85 (set ith HCl) under atmospheric pressure. After 2 hour incubations at 37oC, quantities of MTX-(15N)PG2 formed were measured by LC-MS/MS as described above [56]. FPGS activity is expressed as pmol MTX-PG2 formed per microgram protein per hour (pmol•µg-1•hr-1)

## Data analysis/statistical analysis

Raw luminescence values obtained after readout of MTX drug screens, were normalized to the average of untreated controls (n = 3, 100%) and staurosporin-treated controls (n = 3, 0%), using the formula: (value- 0% control)/(100% control– 0% control)*100%. Resulting percentages of viability were transferred to GraphPad v8 to generate kill curve. Curves were fit using the option 'log inhibitor vs. normalized response–variable slope. IC50s/AUC values were obtained from GraphPad analysis. The change in IC50 values over time was assessed using a Pearson correlation, performed in Graphpad prism, using XY analysis, correlation option. AUC's were compared as previously described. [57, 58]

### Ethics approval and consent to participate

Collection of human tissues was compliant with the guidelines of the European Network of Research Ethics Committees (EUREC) and European and national laws, and written informed consent was obtained from all donors. All donors were > age 18 years. The Biobank Research Ethics Committee of the UMC Utrecht approved the biobanking protocol (12–093 HUB-Cancer).

## Results

### Human wildtype oral mucosa organoids can be used to model MTX-induced cell death *in vitro*

Organoid lines used in this study were derived from tumor-adjacent normal epithelium of patients with head and neck squamous cell carcinoma (S1 Table). Oral mucosa organoids grew as dense structures consisting of epithelial cells that recapitulate the histological organization of the oral mucosal epithelium *in vivo* (Fig 1A and 1B). Keeping in mind their origin and the potential risk of cancer cell contamination, wildtype status of the organoids was confirmed by whole exome sequencing. Using normal epithelial tissue as a reference, a low number of mutations was detected in these cultures (two in N1, none in N2), with no mutations found in common cancer driver genes (S2A Fig, S3 Table). To assess if oral mucosa organoids could serve as a model for MTX toxicity, expression of genes involved in MTX transport, metabolism and toxicity was assessed using quantitative PCR (Fig 1C). In addition, catalytic activity of FPGS (important for MTX retention) was compared with that of a reference human T-cell leukemia cell line CCRF-CEM. FPGS activity was >5 times lower in in oral mucosa organoids than in CCRF-CEM cells (501 versus 2715 pmol MTX-PG$_2$/h/mg protein) (S2B Fig).

Oral mucosa organoids were subsequently exposed to MTX. During MTX treatment, MTX polyglutamates (MTX-PG$_{1-5}$) accumulated intracellularly in oral mucosa organoids in a dose-dependent way (Fig 1D), suggestive of functional MTX metabolism in oral mucosa organoids. Total levels of intracellular MTX-PG differed between organoids derived from different donors, which is in line with the large variation in MTX-PG levels that are detected in patients. MTX-induced cell death was only observed when organoids were cultures in a modified organoid culture medium that contained lower physiological folate levels (Fig 1E), but not the normal organoid culture media (advanced DMEM/F12 containing 6 μM folic acid, 15 μM hypoxanthine and 1.5 μM thymidine). Most likely, when exposed to MTX in this media, these high concentrations prevent MTX toxicity *in vitro* as previously described [59]. [60, 61]. In this low-folate medium, organoids grew at similar speed and showed similar morphology when compared to the previously defined culture medium (S3A and S3B Fig). Therefore, all subsequent drug screens were performed in low folate medium.

### MTX-induced cell death in oral mucosa organoids can be rescued by LV addition, in a timing- and concentration-dependent manner

To assess the role of LV on MTX toxicity, oral mucosa organoids were exposed to a clinically relevant (at levels detected in patient plasma) concentration range of MTX in *in vitro* drug screens (Fig 2A). The drug screen assays showed high technical quality as measured by Z-scores (median 0.72; range 0.31–0.98, S2 Table). To model LV rescue therapy *in vitro*, organoids were exposed to different concentrations of LV at different timepoints after the start of MTX treatment.

Administration of LV resulted in a decrease of MTX-induced cell death in a concentration-dependent manner, that was statistically significant (Fig 2B). Secondly, the extent of LV rescue

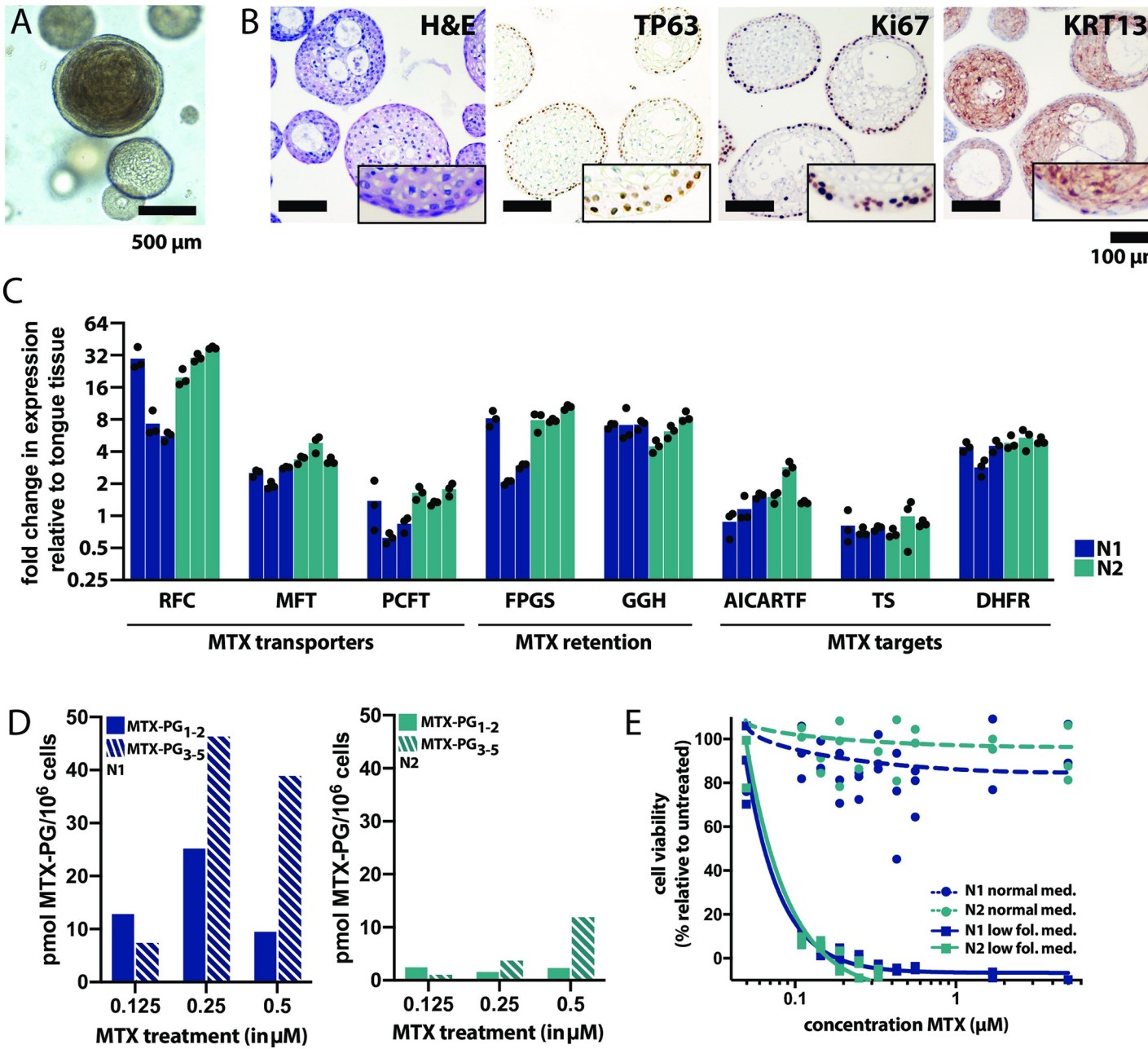

**Fig 1. Human normal oral mucosa organoids can be used to model MTX-induced toxicity *in vitro*.** A. Morhpology of oral mucosa organoids, brightfield microscopy. The cells form round structures with keratinized centers. Scalebar, 500 µm. B. Immunohistochemical stainings performed on paraffin-embedded oral mucosa organoids indicate that organoids resemble the histological characteristics of in vivo oral mucosa epithelium, with a layer of proliferating basal cells (P63/KI67+) on the outside in contact with the BME (an *in vitro* basal lamina mimetic) and more differentiated (KRT13+) cells on the inside of the organoids. Hematoxylin and eosin, TP63 staining, KI67 staining and KRT13 staining are shown from left to right. Scalebar, 100 µm. C. Expression of genes involved in MTX transport, metabolism and toxicity was quantified in low folate medium using quantitative qPCR. Allgenes tested were expressed at detectable levels in oral mucosa organoids. D. Short chain MTX-PG$_{1-2}$ (dashed bars)and long-chain MTX-PG$_{3-5}$ (filled bars) accumulation in two oral mucosa organoid cultures (N1 blue, N2 green) exposed to increasing doses of MTX (0.125, 0.25 and 0.5 µM respectively), detected by UHPLC-MS/MS. E. MTX exposure induces toxicity (cell death) in oral mucosa organoids grown in low folate medium, but not in normal oral mucosa medium that was previously described in Driehuis et al [38]. Organoids were exposed for five days to MTX and viability was quantified relative to untreated organoids. Reproducible killing at physiological doses of MTX was observed in low folate medium only (dashed lines, circle data points), not in normal organoid medium (continuous lines, square data points).

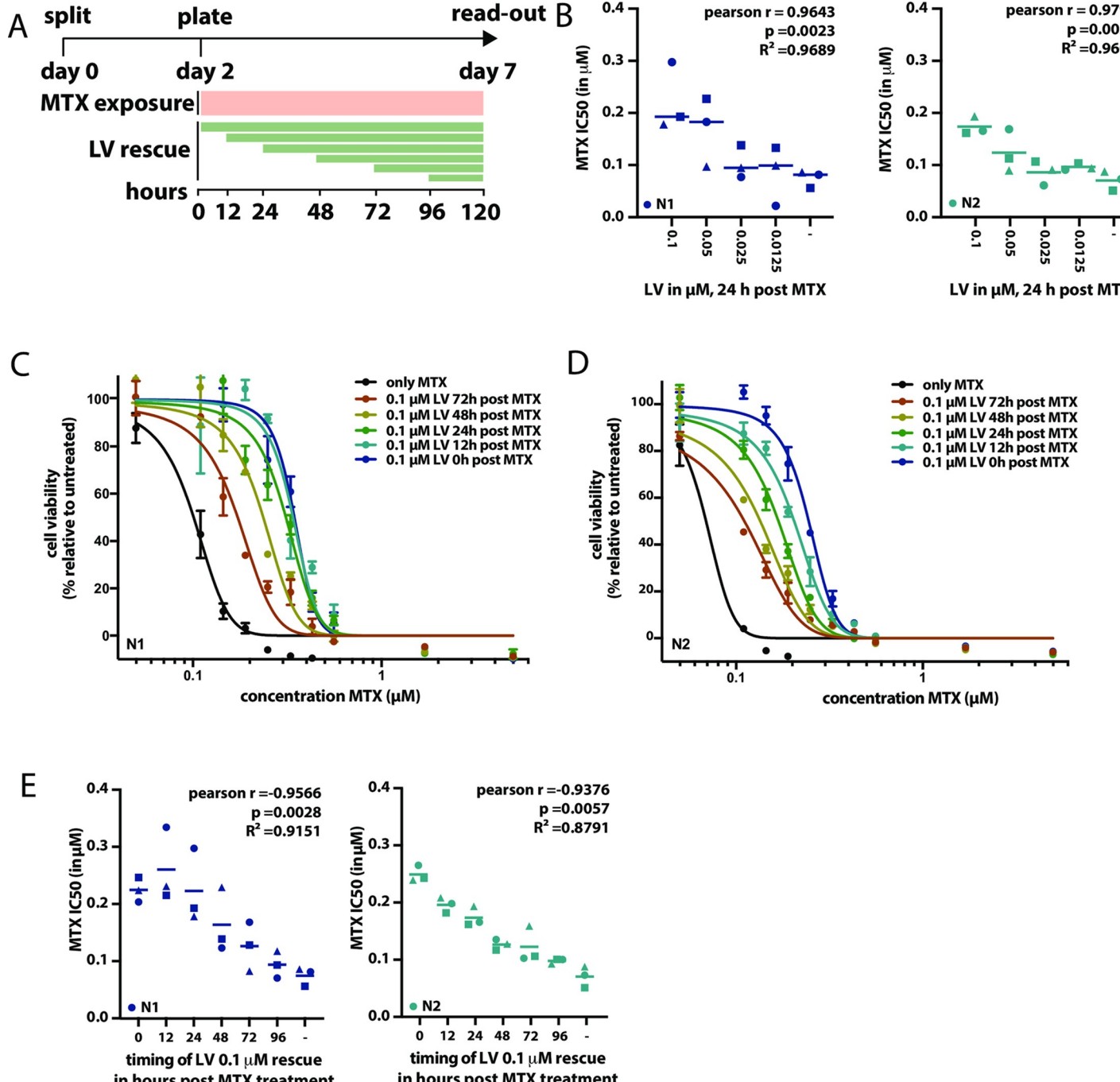

**Fig 2. MTX-induced toxicity in oral mucosa organoids can be rescued by the addition of LV, in both a timing- and dosing-dependent manner. A.** Schematic outline of the experimental set-up applied to assess the effect of LV rescue initiated post MTX exposure toxicity *in vitro*. Organoids were split on day 0, left to recover for two days, subsequently filtered, counted and plated in 384 well format to be exposed to MTX for five days (with or without LV added on variable timepoints after the start of MTX exposure). On day 7, viability readout was performed. As such, organoids were exposed to MTX for a total of five days. **B.** MTX-induced toxicity can be decreased by LV rescue and extent of LV rescue is dose dependent. MTX IC50 values were determined either without LV rescue, and with different LV rescue dosages (0.1 μM, 0.05 μM, 0.025 μM and 0.0125 μM), where rescue was initiated 24 hours post the start of MTX exposure. Each IC50 value is obtained for a 9-point dose titration of MTX, where the effect of each dose is tested in technical triplicate. Symbols indicate the different replicate experiments in which the IC50 were defined for different timepoints of LV rescue. As such, IC50 values indicated with the same symbol at different timepoints, were determined in the same experiment. Blue symbols indicate N1 organoids, turquoise symbols indicate N2 organoids. **C.** Timing of LV rescue influences its effect of MTX-induced toxicity in N1 organoids. Here, MTX kill curves are shown when 0.1 μM LV rescue is initiated either 0, 12, 24, 48, 72 or 96 hours after the start of MTX exposure. **D.** Timing of LV rescue influences its effect of MTX-induced toxicity in N2 organoids. Here, MTX kill curves are shown when 0.1 μM LV rescue is initiated either 0, 12, 24, 48, 72 or 96 hours after the start of MTX exposure. **E.** Effect of PT in N1 (blue symbols) and N2 (turquoise symbols) organoids when combined with LV rescue, initiated at different time-points post start of

MTX exposure. MTX IC50 values are shown on the y-axis. For each timepoint, three IC50 values are show, obtained in three independent experiments. Each IC50 value is obtained for a 9-point dose titration of MTX, where the effect of each dose is tested in technical triplicate. Symbols indicate the different replicate experiments in which the IC50 were defined for different timepoints of LV rescue. As such, IC50 values indicated with the same symbol at different timepoints, were determined in the same experiment.

was dependent on the timing of LV addition; the earlier LV was administrated, the higher the overall cell viability (Fig 2C and 2D). LV administration decreased MTX toxicity up to 72 hours after the start of MTX treatment. A correlation between timing of LV administration and MTX IC50 was observed in both N1 and N2 organoids (Fig 2E). Considering the timing of LV administration varies that can vary per treatment protocol, but is usually initiated at timepoints ranging from 24 to 48 hours after MTX infusion, these results are relevant. As in patients, MTX plasma levels have dropped by 54 hours post infusion, LV administration is rarely continued after this timepoint. Here we observe that, *in vitro*, LV administration still decreases MTX-induced toxicity beyond this timepoint.

## A one day pre-treatment of oral mucosa cells with LV prior to MTX exposure, results in potentiation of the LV rescue effect

In patients, MTX-induced oral mucositis most frequently occurs after the first cycle of HD-MTX [62]. This has resulted in the hypothesis that intracellular LV from previous courses prevents toxicity during subsequent MTX courses. To model this *in vitro*, organoids were exposed to LV one day prior to the start of MTX treatment. At the start of MTX treatment, LV was removed and toxicity was assessed as previously described (Fig 3A). In organoid line N1, LV pre-treatment did not significantly alter the response to MTX (AUC no pre-treatment: 0.8044, AUC pre-treatment 0.8983, t = 1.9169, p = 0.102), although MTX IC50 values marginally increased (Fig 3B and 3C). However, in organoid line N2, a clear rescue effect of the pre-incubation with LV was observed (Fig 3D and 3E), which was also found to be statistically significant (AUC no pre-treatment: 0.7166, AUC pre-treatment 1.034, t = 10.43, p = 0.0005),. LV pre-treatment increased the viability of N2 organoids when exposed to MTX, for all LV rescue timepoints tested here. When pre-treated, the rescue effect of LV rescue administered at 72 hours—later than currently applied in the clinic—resulted in a cell survival similar to a LV rescue that would have been given at 0 hours without this pre-treatment. This suggests that pre-treatment might increase the timeframe in which LV rescue rescues MTX toxicity in oral mucosa cells.

As variable responses to LV pre-treatment were observed in N1 and N2, the effect of pre-treatment was tested in organoids established from three additional donors. In all three cultures, pre-treatment increased oral mucosa cell survival upon exposure to MTX (Fig 3F). Taken together, we conclude that a one day pre-treatment with LV decreases MTX-induced mucosal toxicity in 4/5 tested donors. This implies that LV pre-treatment may reduce the risk of oral mucositis. Regardless, the effect of such a pre-treatment on leukemia cells much be investigated before any claims can be made on clinical testing of such an intervention.

## Effect of MTX and LV therapy on leukemia cell lines

To assess the effect of LV pre-treatment on leukemia cells, both T cell ALL (Jurkat, MOLT16, HSB2) and B cell ALL (Nalm6, REH) cell lines were exposed to MTX, either in the presence or absence of LV pre-treatment (Fig 4A to 4E). Although LV pre-treatment increased MTX IC50 values in the leukemia cell lines tested here, the effect was less pronounced than in oral mucosa cells (Fig 4F).

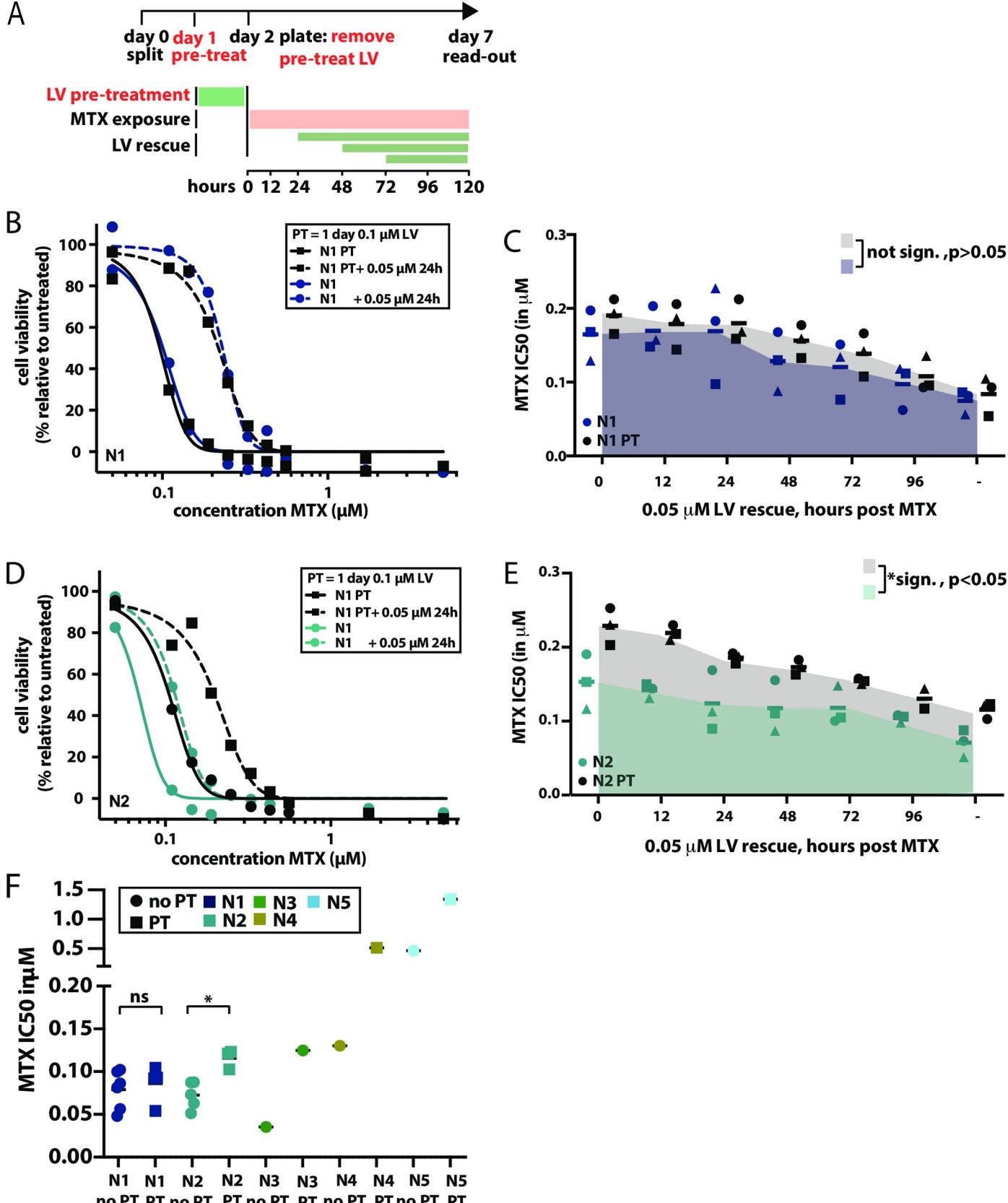

**Fig 3. A one day LV pre-treatment of oral mucosa cells before MTX exposure results in potentiation of the LV rescue effect.** A. Schematic outline of the experimental set-up applied to assess the effect of LV pre-treatment (PT) on MTX toxicity *in vitro*. Deviations from the original experimental set-up used to determine MTX toxicity (see Fig 2A) are shown in red. In short, one day prior to the start of MTX exposure, cells were treated with 0.1 μM LV for 24 hours. Before exposure to MTX, LV was removed. From there one, experimental set-up was identical to that previously described for MTX toxicity. Organoids were exposed to MTX for 5 days, either in the presence or absence of LV rescue, that was commenced at variable times after the start of MTX exposure. B. Effect of a 24 hour 0.1 μM LV PT on MTX toxicity in N1. Cells were exposed to MTX, where MTX exposure was either preceded by a one day 0.01 μM LV PT (black squares) or not (dark blue circles). PT effect was investigated either with (dashed lines) or without (continuous lines) a 0.05 μM LV rescue, applied 24 post treatment. Toxicity was determined using a 9-point dose titration of MTX, where the effect of each dose is tested in technical triplicates. Pearson correlation coefficient was determined in GraphPad to evaluate the correlation between dose of LV rescue and MTX IC50. C. Effect of PT in N1 organoids when combined with LV rescue, initiated at different time-points post start of MTX exposure. MTX IC50 values are shown on the y-axis. For each timepoint, three IC50 values are show, obtained in three independent experiments. Each IC50 value is obtained for a 9-point dose titration of MTX, where the effect of each dose is tested in technical triplicate. Symbols indicate the different replicate experiments in which the IC50 were defined for different timepoints of LV rescue. As such, IC50 values indicated with the same symbol at different timepoints, were determined in the same experiment. D. Effect of a 24 hour 0.1 μM LV PT on MTX toxicity in N2. Cells were exposed to MTX, where MTX exposure was either preceded by a one day 0.1 μM LV PT (black squares) or not (turquoise circles). PT effect was investigated either with (dashed lines) or without (continuous lines) a 0.05 μM LV rescue, applied 24 post treatment. Toxicity was determined using a 9-point dose titration of MTX, where the effect of each dose is tested in technical triplicates. E. Effect of PT in N2 organoids when combined with LV rescue, initiated at different time-points post start of MTX exposure. MTX IC50 values are shown on the y-axis. For each timepoint, three IC50 values are show, obtained in three independent experiments. Each IC50 value is obtained for a 9-point dose titration of MTX, where the effect of each dose is tested in technical triplicate. Symbols indicate the different replicate experiments in which the IC50 were defined for different timepoints of LV rescue. As such, IC50 values indicated with the same symbol at different timepoints, were determined in the same experiment. Pearson correlation coefficient was determined in GraphPad to evaluate the correlation between timing of LV rescue and MTX IC50. F. Effect of LV pre-treatment (PT) on viability of N1, N2, N3, N4 and N5 organoids, respectively, all derived from different donors. Squares indicate PT conditions, circles indicate cells that did not receive PT. In 4/5 organoid cultures (all organoid cultures apart from N1), an increase in MTX IC50 value can be observed in response to pre-treatment, indicating that a 24 hour 0.1 μM LV PT reduces MTX toxicity in oral mucosa cells.. Here, IC50 values are shown when no LV rescue is performed. For N1 and N2, experimetns were not only performed in technical triplicate, but also biological triplicate. Here, t-tests were performed to confirm that in N2, but not N1, a statistical significant effect of LV preptreatment could be observed on MTX IC50 values.

To estimate the effect of LV pre-treatment when administered systemically, we compared the effect of this treatment on both oral mucosa cells and leukemia cells at a concentration of 0.1 μM MTX (a level reached during plasma measurements of all patients) [62]. At this MTX dose, LV pre-treatment decreases the toxicity of MTX on oral mucosa cells, but does not influence the effect of MTX on 4/5 leukemia cells (Fig 4G). T-ALL cell line Jurkat did show decreased MTX toxicity when exposed to LV pre-treatment. Indeed, it has been previously reported that T-cell ALL is less sensitive than B-cell ALL [63, 64]. Indeed, MTX-PG levels in two B-ALL leukemia cell lines (REH; Nalm6) were around 3-fold higher (S2C Fig) when compared to MTX-PG levels in oral mucosa organoids (S2D Fig). MTX-PG levels in two T-ALL leukemia cell lines (Jurkat; HSB2) differed with high MTX-PG levels in Jurkat cells (~3-fold higher than organoids) and low MTX-PG levels in HSB-2 cells (same range as organoids).

Taken together, we conclude that LV pre-treatment decreases MTX-induced cell death of leukemia cells *in vitro*. However, as leukemia cells are much more sensitive to MTX than oral mucosa cells, the effect of this pre-treatment might be mitigated at the MTX levels reached during HD-MTX infusions. Regardless, the effect of LV pre-treatment should be explored with caution, but might prove to be an effective approach to decrease the severity and frequency of MTX-indicued mucositis. A local LV application is a feasible alternative, since it would likely not interfere with the systemic MTX effect on leukemia cells and may therefore be a safer approach to reduce the risk of mucositis.

## Discussion

Using a 3D *in vitro* model of primary human oral mucosa organoids, we show donor-dependent MTX-induced cell death in wildtype human oral mucosa cells. To our knowledge, this is the first *in vitro* model (based on the model described by Driehuis et al [38] with modification in media composition) to assess the effect of MTX on proliferating, wildtype oral mucosa epithelial cells. Using this system, we show that administration of LV at dosages detected in patient plasma, reduces MTX-induced cell death in a concentration- and time-dependent

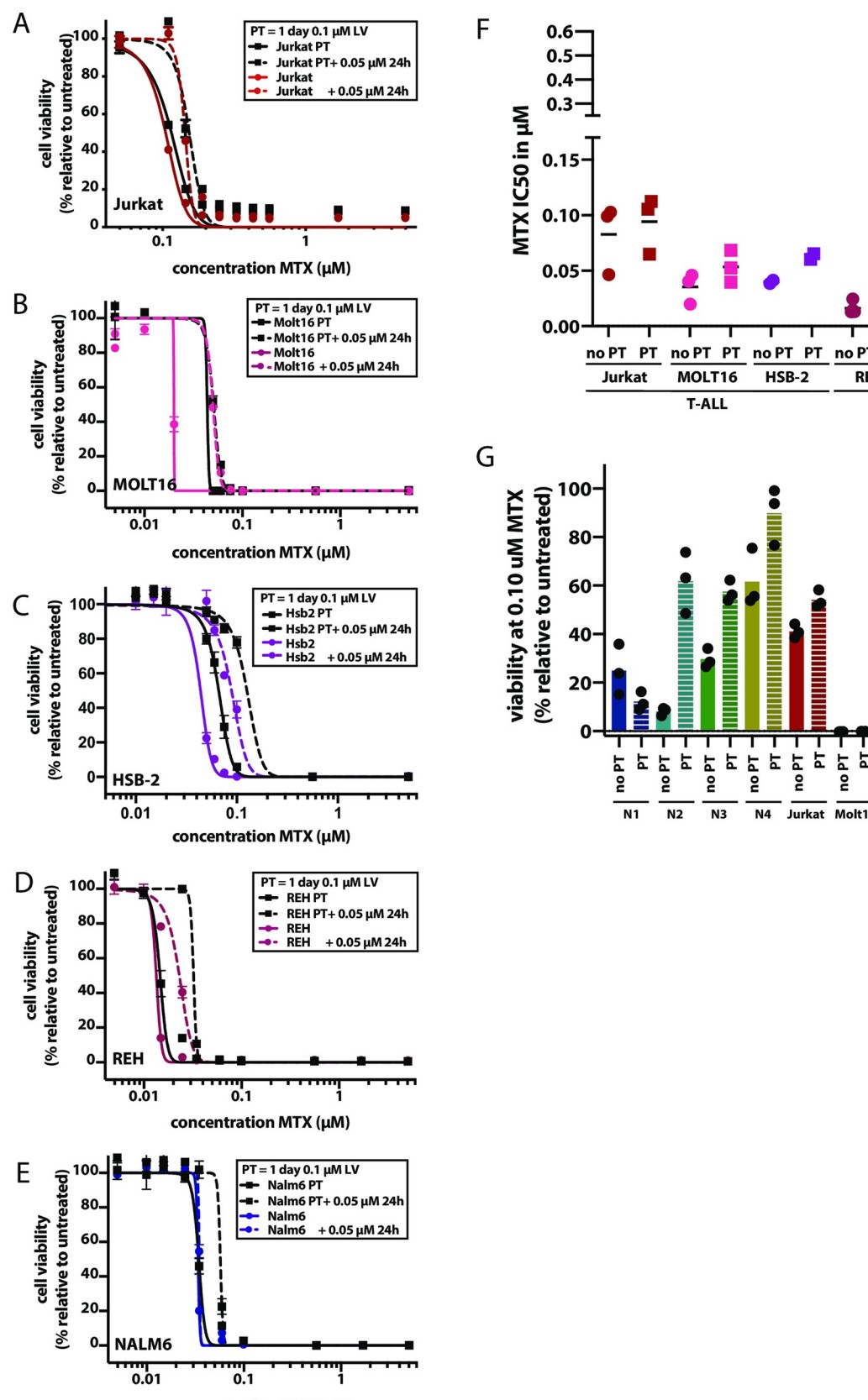

**Fig 4. LV pre-treatment influences MTX toxicity in leukemia cell lines.** A to E. Effect of LV pre-treatment (PT) on viability of T-ALL and B-ALL leukemia cell lines. For each cell line, cells were exposed to MTX, where MTX exposure was either preceded by a one day 0.01 μM LV PT (black squares) or not (colored circles, color indicating the cell line tested). PT effect was investigated either with (dashed lines) or without (continuous lines) a 0.05 μM LV rescue, applied 24 post treatment. A. Jurkat, dark red. B. MOLT16, fuchsia. C. HSB-2, purple. D. REH, magenta. E. Nalm-6, violet. F. MTX IC50 values detected for leukemia cell lines when exposed to MTX, either preceded by a0.01 μ M LVPT (squares),or not (circles). For each cell line, three IC50 values are show, obtained in three independent experiments. Each IC50 value is obtained for a 9-point dose titration of MTX, where the effect of each dose is tested in technical triplicate. In all cases, an increase in MTX IC50 value can be observed in response to pre-treatment. Here, IC50 values are shown when no LV rescue is performed. Different colors indicate the different cell lines, colors are similar to those used in A-E. G. Viability of cells exposed for 5 days to 0.1 μM MTX, either preceded by 0.01 μM LV PT (dashed bars) or not. Viability was tested in technical triplicate, where the average viability is shown for three independent experiments. Different colors indicate the different cell lines, colors are similar to those used in A-E.

manner and that MTX-induced toxicity shows variability between cells derived from different individuals.

In patients, LV infusion after HD-MTX is usually initiated 36 or 42 hours after HD-MTX, although in some protocols, LV is already applied at 24 hours post MTX infusion. LV rescue therapy is often not administered beyond the timepoint of 54 hours after start of MTX. [65] Based on our findings, we hypothesize that both an earlier start and longer continuation of LV rescue might reduce the incidence or extent of MTX-induced mucositis. However, since the model presented here only studies the effect of such interventions on either mucosal cells or leukemia cell lines, these results need to be validated (for example in mouse models) before they can be clinically tested. Alternatively, a local application of LV one day prior to the start of MTX administration might be an alternative approach to reduce mucositis in pediatric leukemia patients. A local application of LV is currently already applied in some clinics when patients present with oral mucositis after HD-MTX.

To assess the effect of LV pre-treatment on leukemia cells, leukemia-derived cell lines were exposed to the same pre-treatment to study the effect of this intervention on MTX toxicity. Although present, the effect of pre-treatment and LV rescue was less pronounced in leukemia cells than in oral mucosa cells. Differences in response to LV between leukemia cells and healthy cells have been observed before. Several pre-clinical studies showed this selective mechanism of action for MTX and LV might be due to the fact that high MTX-PG levels accumulate in leukemia cell lines compared to normal intestinal and bone marrow precursor cells [14–22]. Also in oral mucosa organoids we found that MTX-PG levels and FPGS activity are lower than in the leukemia cell lines tested here. Taken together, these observations support the fact that primary patient-derived ALL cells (especially B-ALL when compared to T-ALL) are sensitive to MTX due to a high proliferation rate and high FPGS activity and thus might be less affected by LV than oral mucosa cells [22]. However, it is important to test clinical safety of the described interventions in order not to decrease anti-leukemic activity of MTX.

In clinics, only a subset of patients presents with mucositis, suggesting some patients are more sensitive to MTX treatment than others, or might respond better to LV rescue treatment. This is in line with our findings, where organoid cultures derived from different patients, show variable responses to both MTX and LV exposure *in vitro*. Due to this variable response, more personalized approaches to decrease the risk of MTX-induced mucositis are urgently needed and, in some clinics, already explored. Although beyond the scope of this work, oral mucosa organoids create the opportunity to study the molecular differences between sensitive and resistant cultures. If these inter-patient differences are a consequence of local rather than systemic differences in responses to MTX/LV, such comparisons could help to identify patients at risk, or contribute to the development of alternative treatment strategies to decrease the incidence of oral mucositis.

Here we only study MTX-induced cell death as a model for mucositis. Importantly, oral mucositis is a complex process of which cell death is only one of the hallmarks [8, 66]. In 2004, Sonis *et al*. proposed a more complicated model of oral mucositis, in which the generation of

Reactive Oxygen Species (ROS) and or pro-inflammatory cytokines were also hallmarks of the clinical phenotype in addition to therapy-induced cell death [8]. Furthermore, it has been suggested that the bacterial microbiome might play a role in developing oral mucositis. It would be of interest to study the effect of these factors in future models [8]. Co-cultures of organoids and immune cells or organoids and bacteria have been described [67, 68]. Therefore, this model holds the potential to be extended to investigate the contribution of these factors in oral mucositis.

By making minor adaptations to the culture media that was previously described to culture oral mucosa organoids [38], we have shown that organoids can be used to study MTX-toxicity in vitro. These modifications can be applied to other organoid cultures such as kidney or liver organoids [69, 70]. As such, these models can perhaps be used to study MTX-toxicity that is observed in other organs than the oral mucosa.

## Conclusion

Although applied in clinic for many years, the effect of LV rescue therapy to reduce oral mucositis after MTX treatment has not been shown in representative models before. Here, we report the use of normal human oral mucosa organoids that recapitulate functional and histological characteristics of this epithelium to study the potential of LV to reduce MTX-induced toxicity. Oral mucosa organoids show sensitivity to clinically relevant doses of MTX in vitro. MTX-induced toxicity could be reduced by the addition of LV after the start of MTX treatment. The extent of LV-rescue is concentration- and time- dependent and differs between organoids derived from different donors, fitting with what is observed in the clinic. Using this system, we find a pre-exposure ('pre-treatment') with LV before MTX treatment significantly potentiates the effect of LV rescue. The effect of this pre-treatment is present, but less pronounced in leukemia cells. Taken together, our findings support the LV rescue protocols currently applied in the clinic and, moreover, highlight the potential of this model to study the effect of modifications of the currently applied clinical regimens, such as a LV pre-treatment, on proliferation wildtype oral mucosa cells *in vitro*.

## Supporting information

**S1 Fig. Mechanism of action of MTX treatment.** Leucovorin (5-formylTHF) is represented in bold / italic. MTX enters the cell mainly through the Reduced Folate Carrier 1 (RFC1), Proton Coupled Folate Transporter (PCFT), Membrane Folate Transporters (MFR) or by passive diffusion through the cell membrane. While circulating, MTX contains one polyglutamate group (MTX-PG$_1$). Once inside the cell, MTX is polyglutamated by Folylpolyglutamate Synthetase (FPGS) with up to seven polyglutamate groups. Long-chain MTX-PG's (MTX-PG$_{4-7}$) cannot be transported out of the cell before de-polyglutamation by Gamma-Glutamyl Hydrolase (GGH). Short-chain MTX-PG's (MTX-PG$_{1-3}$) will be actively transported out of the cell by ABCC1-4, ABCB1 and ABCG2 transporters. MTX is cytotoxic as it impairs purine- and pyrimidine synthesis by inhibiting the enzymes Dihydrofolate Reductase (DHFR) and Thymidylate Synthase (TYMS). Abbreviations: ABCB1—ATP Binding Cassette Subfamily B Member 1; ABCC1-4—ATP Binding Cassette Subfamily C Member 1–4; ABCG2—ATP Binding Cassette Subfamily G Member 2; DHFR–Dihydrofolate Reductase; FPGS–Folylpolyglutamate Synthetase; GGH–Gamma-Glutamyl Hydrolase; MFR–Membrane Folate Transporter; MTHFR—Methylene tetrahydrofolate reductase; MTHFD1—Methylenetetrahydrofolate Dehydrogenase, Cyclohydrolase And Formyltetrahydrofolate Synthetase 1; PCFT–Proton-Coupled Folate Transporter; RFC1 –Reduced Folate Carrier; SHMT—Serine hydroxymethyl-transferase; TS–Thymidylate Synthase.
(PDF)

**S2 Fig. Characterization of oral mucosa organoids and the effect of pretreatment on intracellular MTX-PG levels.** A. Oral mucosa organoids are derived of human normal cells, and not cancer cells. Number of mutations detected by whole exome sequencing in the healthy oral mucosa organoids used in this study, and their corresponding tumor organoids. Mutational load is low (2 for N1, 0 for T1), especially when compared to the tumor organoids. B. FPGS activity (in pmol MTX-PG$_2$/h/mg) in organoid line versus CCRF-CEM reference leukemia cell line. C. Effect of PT on MTX-PG levels in oral mucosa organoid lines derived from two different donors. D. Effect of PT on MTX-PG levels in two B-ALL and two T-ALL cell lines.
(PDF)

**S3 Fig. Organoid cultures retain their morphology and growth speed when grown in folate deprived medium.** A. Brightfield microscopy images of organoid line N1 and N2, when grown in either complete medium, or folate deprived medium. Scalebar, 500 μm. B. Growth speed of organoid cultures in both media tested. Growth was assessed by collection of cell pellets at day 0, 3, 5, 7, 10 and 14. Cell number was assessed by cell titer glow and values were made relative to day 0. C. Quantitative PCR assessing expression of genes relevant for methotrexate metabolism. Experiment was performed in triplicate, results of all three experiments are shown here.
(PDF)

**S4 Fig. Technical details of drugscreen performed in this study.** A. Schematic layout of a drug screen plate as used in this study. The gradient of MTX is depicted using a color gradient (red indicates high concentration, green indicates low concentration). Here, the MTX concentrations used for organoids are depicted. Each concentration is tested in technical triplicate. Different blocks receive LV rescue at different timepoints after the start of MTX treatment, as indicated. Staurosporine treated wells are used as positive controls and are set to 0% viability, wells only receiving drug solvent are used is negative controls, and are set to 100% viability. B. Brightfield microscopy images showing the morphology of N1 organoids in drug screening plates on the day of readout. C. Brightfield microscopy images showing the morphology of N2 organoids in drug screening plates on the day of readout.
(PDF)

**S1 Table. Clinical information of patients. Relevant clinical information is given on the patient that participated in this study, and form whose tissue organoids were derived.**
(PDF)

**S2 Table. Z-scores of drug screens performed in this study.**
(PDF)

**S3 Table. Comparison of mutations detected by WES in matching normal and tumor organoid lines.** All mutation detected in organoid line N1, T1, N2 and T2 are shown. Here, normal tissue was used as a reference.
(XLSX)

**S4 Table. Sequences of primers used for quantitative PCR.** 5' to 3' sequences of primers used to assess gene expression by quantitative PCR in this study.
(PDF)

## Acknowledgments

We would like to acknowledge Onno Kranenburg, Anneta Brousali, Petra van der Groep, Alexander Constantinides and Anne Snelting of the Utrecht Platform for Organoid

Technology (U-PORT; UMC Utrecht) for patient inclusion and tissue acquisition. We thank HUB Organoids for help with regulatory affairs regarding informed consent. We would like to thank Sacha Spelier for her help with organoid experiments. We acknowledge Eduard Struys for technical support in FPGS activity and MTX-PG level measurements. We thank the groups of Monique den Boer and Jules Meijerink at the Princess Maxima Center for providing us with leukemia cell lines.

## Author Contributions

**Conceptualization:** E. Driehuis, N. Oosterom, S. G. Heil, G. Jansen, M. M. van den Heuvel-Eibrink.

**Data curation:** E. Driehuis, N. Oosterom.

**Formal analysis:** E. Driehuis, N. Oosterom, I. B. Muller, M. Lin, S. Kolders.

**Funding acquisition:** E. Driehuis.

**Investigation:** E. Driehuis, N. Oosterom, S. G. Heil, I. B. Muller, M. Lin, S. Kolders, G. Jansen, R. de Jonge.

**Methodology:** E. Driehuis, N. Oosterom, S. G. Heil, I. B. Muller, M. Lin, G. Jansen, R. de Jonge.

**Project administration:** E. Driehuis.

**Resources:** H. Clevers, M. M. van den Heuvel-Eibrink.

**Software:** E. Driehuis.

**Supervision:** G. Jansen, R. de Jonge, R. Pieters, H. Clevers, M. M. van den Heuvel-Eibrink.

**Validation:** E. Driehuis, N. Oosterom, R. de Jonge.

**Visualization:** E. Driehuis, N. Oosterom.

**Writing – original draft:** E. Driehuis, N. Oosterom.

**Writing – review & editing:** E. Driehuis, N. Oosterom, S. G. Heil, I. B. Muller, M. Lin, S. Kolders, G. Jansen, R. de Jonge, R. Pieters, H. Clevers, M. M. van den Heuvel-Eibrink.

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
