## [Decision Letter · Decision Letter 0]

15 Jan 2020

PONE-D-19-32380

Patient-derived oral mucosa organoids as an in vitro model for methotrexate induced toxicity in pediatric acute lymphoblastic leukemia

PLOS ONE

Dear Ms Oosterom,

Thank you for submitting your manuscript to PLOS ONE. After careful consideration, we feel that it has merit but does not fully meet PLOS ONE’s publication criteria as it currently stands. Therefore, we invite you to submit a revised version of the manuscript that addresses the points raised during the review process.

Please address all the points raised by the reviewers carefully point by point.

We would appreciate receiving your revised manuscript by Feb 29 2020 11:59PM. To enhance the reproducibility of your results, we recommend that if applicable you deposit your laboratory protocols in protocols.io, where a protocol can be assigned its own identifier (DOI) such that it can be cited independently in the future. For instructions see: http://journals.plos.org/plosone/s/submission-guidelines#loc-laboratory-protocols

We look forward to receiving your revised manuscript.

Kind regards,

Obul Reddy Bandapalli, MSc, PhD

Academic Editor

PLOS ONE

Journal Requirements:

2. Please provide additional details regarding participant consent. In the ethics statement in the Methods and online submission information, please ensure that you have specified what type of consent you obtained (for instance, written or verbal). If your study included minors under age 18, state whether you obtained consent from parents or guardians. If the need for consent was waived by the ethics committee, please include this information.

3. We noticed you have some minor occurrence(s) of overlapping text with the following previous publication(s), which needs to be addressed:

https://doi.org/10.1158/2159-8290.CD-18-1522

https://doi.org/10.1097/FTD.0000000000000638

In your revision ensure you cite all your sources (including your own works), and quote or rephrase any duplicated text outside the Methods section. Further consideration is dependent on these concerns being addressed.

4. We note that this submission reports a functional enzymological study with kinetic and thermodynamic data. The reporting of these data should include the temperature, pH and pressure, as well as the identity of the catalyst and its origins, the method of preparation, criteria for purity and assay conditions. We recommend that you refer to the Standards for Reporting Enzymology Data (STRENDA) of the Beilstein Institut for details regarding the adequate description of experimental conditions and reporting of enzyme activity data: https://www.beilstein-strenda-db.org/strenda/public/guidelines.xhtml. Please note that the Beilstein Institut’s STRENDA database automatically checks manuscript data for guideline compliance, as well as making them publicly available after publication and assigning them a specific DOI number for reference and tracking purposes. If you obtain a STRENDA Registry number (SRN) and PDF containing all your functional enzymology data, please include these as Supplementary files.

5. Please report in your Methods a paragraph on immunohistochemistry staining.

6. In your Methods section, please give the sources of any cell lines used in your study.

7. We note that you have a patent relating to material pertinent to this article. Please provide an amended statement of Competing Interests to declare this patent (with details including name and number), along with any other relevant declarations relating to employment, consultancy, patents, products in development or modified products etc. Please confirm that this does not alter your adherence to all PLOS ONE policies on sharing data and materials, as detailed online in our guide for authors http://journals.plos.org/plosone/s/competing-interests by including the following statement: "This does not alter our adherence to  PLOS ONE policies on sharing data and materials.” If there are restrictions on sharing of data and/or materials, please state these. Please note that we cannot proceed with consideration of your article until this information has been declared.

8. Please upload a copy of Supporting Information Table S3 which you refer to in your text on page 13.

Reviewers' comments:

Reviewer's Responses to Questions

**Comments to the Author**

1. Is the manuscript technically sound, and do the data support the conclusions?

Reviewer #1: Yes

Reviewer #2: No

Reviewer #3: Yes

2. Has the statistical analysis been performed appropriately and rigorously? 

Reviewer #1: Yes

Reviewer #2: No

Reviewer #3: Yes

3. Have the authors made all data underlying the findings in their manuscript fully available?

Reviewer #1: Yes

Reviewer #2: Yes

Reviewer #3: Yes

4. Is the manuscript presented in an intelligible fashion and written in standard English?

Reviewer #1: Yes

Reviewer #2: Yes

Reviewer #3: Yes

5. Review Comments to the Author

Reviewer #1: In the original research article entitled, “Patient-derived oral mucosa organoids as an in vitro model for methotrexate induced toxicity in pediatric acute lymphoblastic leukemia“, Else Driehuis and colleagues describe a novel assay to assess methotrexate-induced toxicity in human squamous cells. They have established a protocol to grow human wildtype oral mucosa organoids. These three-dimensional structures can be maintained in culture, do not require immortalization, and recapitulate the multilayered composition of the epithelial lining of the oral mucosa. They have utilized their co-culture model to assess the effects of leucovorin rescue on methotrexate-induced squamous cell damage as functions of timing and concentration. The paper is interesting and well-presented, and the experiments provide compelling new information about the use of leucovorin as a rescue agent for methotrexate. Some limitations exist, which will be detailed in the sections that follow.

Major Considerations:

1. In general, methotrexate toxicity is not constrained to oral mucositis alone. Hepatic and hematological toxicities are also dose-limiting factors. The paper does not address these issues, which are common limitations in dose-intensified therapies. Moreover, leucovoroin is a highly soluble drug, and pretreatment may have unintended complications in over-rescuing methotrexate (over-rescue literature should be cited). The paper should be revised to include these limitations, and statement about the use of leucovorin pre-treatment should be modified to reflect the potential down-side of changing rescue strategies.

2. In clinical practice, methotrexate-induced oral mucositis can be somewhat mitigated by the use of oral bicarbonate rinses. Was this condition used as a control variable in the experiments that were performed? If so, what was the relative impact compared to conditions that used leucovorin pre-treatment?

3. The Discussion section could be shorter, since some of the same points are re-stated in the Conclusion paragraph.

Lesser Points:

1. At line 300, “As, clinically….” appears to be missing a word.

2. The authors should define IC50 at its first use.

3. The legend for Figure 4 could be more clearly written to help the reader. understand what effects LV rescue have on MTX toxicity in leukemia cell lines.

Reviewer #2: The authors report the use of oral mucosal organoids from adjacent normal tissue of patients who underwent surgery for head and neck squamous cell carcinoma. The development of these organoids is interesting, but while this manuscript gives the impression that these are new organoids, these are in fact previously published (ref #34 cited by the authors; this should have been made clear). The authors use these organoids to study oral toxicity of MTX and its rescue using leucovorin. There are a number of problems with this manuscript, at least two of which rise to the level of fatal flaws: Statistical analyses to support the differences claimed are lacking entirely, and critique #4 below is viewed as a completely unacceptable attempt to mislead the reviewers/readers (this may be a typo, but then this raises questions about rigor elsewhere in this work).

Critiques:

1. The drug unit labels are unnecessarily confusing on the viability plots (i.e., Fig. 1E, 2C, etc). It is difficult for most people to figure out what -0.5 log10 uM converts to, and this is the approximate effect size (based in change in IC50) induced by leucovorin treatment. It is fine to show the data on log scale, but it would be much easier for the doses to be shown as 1 uM, 0.1 uM, 0.01 uM, etc.

2. Why are the number of replicates from different timepoints/conditions so different in many experiments? For example Fig 1E, 2B, 2E, and several others. This gives the strong impression that the data shown are an amalgamation of different experiments performed at different times. If the experiments were performed simultaneously, this should be clearly stated in the methods. If the data are a mix of experiments at same vs different time points, all of the experiments should be repeated at one time point with the same number of replicates per condition. It is a good idea to repeat experiments to ensure reproducibility, but results of only one representative experiment should be shown.

3. Are the changes in IC50s induced by leucovorin in Fig 1E, 3C, 3E and 3F statistically significant? Significance of all relevant comparisons should be assessed, with appropriate correction for multiple hypothesis testing. The lack of statistical analyses to support the conclusions is a major flaw throughout this paper.

4. LV pre-treatment also protects leukemia cells. The authors claim that this is less than in normal oral mucosa, but again there is no statistical evidence to support that this is a significant difference. FURTHERMORE, IT IS GLARING THAT THE PRE-TREATMENT OF LEUKEMIA CELLS IS DONE WITH A DOSE OF LEUKOCORIN 10-FOLD LOWER THAN THAT OF THE ORGANOIDS!!! As indicated on the plots of Fig 3B, PT = 0.1 uM in organoids, and Fig 4, 0.01 uM leukemia cells. Is it any surprise that there is less rescue with a lower dose of leucovorin?? This is viewed as a completely unacceptable attempt to mislead the reviewers and readers, and in the opinion of this reviewer this alone is grounds for rejection of the paper. Unless perhaps this is a typo, but then this raises concerns about rigor elsewhere in this manuscript.

Reviewer #3: This is an original article that describes a series of experiments designed to explore the impact of methotrexate and leucovorin on the viability of cells in human oral mucosa organoids, as a model for MTX induced mucositis, an important side effect that arises in some patients in the course of treatment for childhood ALL. The authors report that, as expected, mucosal cells are killed by methotrexate in a time and concentration dependent manner. Additionally, mucosal cells also recapitulated the expected behavior of becoming more resistant to MTX-induced death upon co-incubation with leucovorin, an effect dependent upon the timing of addition as well as to the concentration of the two inhibitors.

The experiments are very well described and in sufficient detail, which should enable others to reproduce and extend these findings. Although the results are completely in accord with our understanding of the mechanisms of MTX and leucovorin and the clinical experiences with them, the manuscript provides clarity and a solid basis to support the investigators’ claim that this is a reliable model for identifying factors that modulate mucositis development in ALL patients. As pointed out by the authors, mucositis in vivo may involve additional mechanisms such as the microbiome and/or the immune system, factors absent from their model. Nonetheless, the ability to test cells from different individuals with more or less sensitivity to MTX could significantly advance our understanding of genetic factors that pre-dispose to this toxicity, above and beyond what is already known about MTX metabolism.

Mucositis occurs not infrequently during ALL treatment, and so this is an important clinical aspect of ALL therapy. The authors comments on the details of MTX and leucovorin treatment in childhood ALL are appropriate, and I agree with their note that earlier treatment with leucovorin (including pre-treatment) may impair the disease responses to MTX, so should be considered with care. Nonetheless, the idea of topical application of leucovorin to the oral mucosa is a good one, and could improve toxicity with minimal negative impact on overall cure rates.

The article is well written and the narrative flows well. The manuscript would benefit from a few additions. First of all, the authors should clarify the state of the mucosal cells during the experiment. Are they isolated as single cells, which seems to be the case since the authors report they trypsinize them just prior to adding drugs? Or do they have time to assemble into the 3d organoids, as mentioned in the title? Secondly, there is little or no discussion of why this system is superior to testing of individual isolated cell cultures, and so it would be good to comment on that. Do the authors think that penetration of drugs into the interior of 3d organoids is an important factor?

6. PLOS authors have the option to publish the peer review history of their article (what does this mean?). If published, this will include your full peer review and any attached files.

Reviewer #1: No

Reviewer #2: No

Reviewer #3: No

---

## [Author Response · Author response to Decision Letter 0]

3 Mar 2020

To: Obul Reddy Bandapalli (MSc, PhD), Academic Editor PLOS ONE

Subject: rebuttal Ms. Ref. No. PONE-D-19-32380

Dear Dr. Bandapalli,

We are pleased to hereby resubmit our revised manuscript ‘Patient-derived oral mucosa organoids as an in vitro model for methotrexate induced toxicity in pediatric acute lymphoblastic leukemia’. 

We thank you and the reviewers for the time and effort put into reviewing our work, and appreciate the constructive criticism. We have addressed each the concerns raised by the reviewers, and give a point-by-point rebuttal below.

Major changes in the revised manuscript include: 

- The introduction of a Log-axis in all the figures, which makes them easier to interpret for the reader. 

- Statistical analysis to further support the biological differences observed between different experimental groups in our experiments. 

- Rewriting of the discussion section to have less overlap with the conclusion. 

- Rewriting of the Figure legends. 

With these modifications, we feel that our manuscript has improved, and we hope you would consider our study for publication in Plos One. 

Best regards, 

Prof.Dr. Marry M. van den Heuvel-Eibrink

 

Editor

We have checked the style templates and the manuscript now fits the requirements. 

2. Please provide additional details regarding participant consent. In the ethics statement in the Methods and online submission information, please ensure that you have specified what type of consent you obtained (for instance, written or verbal). If your study included minors under age 18, state whether you obtained consent from parents or guardians. If the need for consent was waived by the ethics committee, please include this information.

We have made the ethics statement more complete as following:

“Collection of human tissues was compliant with the guidelines of the European Network of Research Ethics Committees (EUREC) and European and national laws, and written informed consent was obtained from all donors. All donors were > age 18 years. The Biobank Research Ethics Committee of the UMC Utrecht approved the biobanking protocol (12-093 HUB-Cancer).” 

3. We noticed you have some minor occurrence(s) of overlapping text with the following previous publication(s), which needs to be addressed:

https://doi.org/10.1158/2159-8290.CD-18-1522

https://doi.org/10.1097/FTD.0000000000000638

In your revision ensure you cite all your sources (including your own works), and quote or rephrase any duplicated text outside the Methods section. Further consideration is dependent on these concerns being addressed.

We have referred to the publications mentioned above. 

4. We note that this submission reports a functional enzymological study with kinetic and thermodynamic data. The reporting of these data should include the temperature, pH and pressure, as well as the identity of the catalyst and its origins, the method of preparation, criteria for purity and assay conditions. We recommend that you refer to the Standards for Reporting Enzymology Data (STRENDA) of the Beilstein Institut for details regarding the adequate description of experimental conditions and reporting of enzyme activity data: https://www.beilstein-strenda-db.org/strenda/public/guidelines.xhtml. Please note that the Beilstein Institut’s STRENDA database automatically checks manuscript data for guideline compliance, as well as making them publicly available after publication and assigning them a specific DOI number for reference and tracking purposes. If you obtain a STRENDA Registry number (SRN) and PDF containing all your functional enzymology data, please include these as Supplementary files.

We have made changes accordingly:

“FPGS catalytic activity analysis in organoids and leukemic cell lines was performed essentially as described by Muller et al (2019, PMID 31008996). In short, FPGS protein was isolated from organoids and cell lines by sonication (Sonoplus Mini 20; Bandelin, Berlin, Germany) on ice for 2 · 10 seconds (10 second intervals with 90% amplitude and 30-second intervals between samples) in FPGS extraction buffer (50 mM Tris, 20 mM KCl, 10 mM MgCl2, and 5 mM DTT, pH 7.4, 4oC, in MilliQ (MilliQ Advantage A10; Merck Millipore, Burlington, MA, USA), followed by centrifugation in an Eppendorf centrifuge (12.000 • g, 15 minutes, 4°C). FPGS-mediated conversion of MTX-PG1 to MTX-PG2 is determined in cell extracts (10-200 µg protein) in a total volume of 250 µL containing final concentrations of 100 mM Tris, 20 mM KCl, 20 mM MgCl2, 10 mM DTT, 10 mM ATP, 250 mM MTX-PG1, and 4 mM 15N-labeled L glutamatic acid (Sigma-Aldrich, cat no. 332143-100MG) at a pH of 8.85 (set ith HCl) under atmospheric pressure. After 2 hour incubations at 37oC, quantities of MTX-(15N)PG2 formed were measured by LC-MS/MS as described above (Muller et al (2019, PMID 31008996). FPGS activity is expressed as pmol MTX-PG2 formed per microgram protein per hour (pmol•µg-1•hr-1).”

5. Please report in your Methods a paragraph on immunohistochemistry staining.

The following section was added in the revised manuscript: 

Immunohistochemistry

Tissues or organoids were fixed in 4% paraformaldehyde overnight, dehydrated, and 

 embedded in paraffin. Sections were subjected to H&E as well as IHC staining, using the 

 antibodies shown in the table below. Stainings were performed at the pathology department 

 of the UMCU (Utrecht, the Netherlands).

Protein supplier Order nr. Host species Clone Lot nr. Dilution Antigen retrieval method

TP63 Abcam AB735 Mouse 4AB AB735 1:800 Citrate

KI67 Monosan MONX10293 Mouse MM1 10293 1:2000 Citrate autoclave

KRT13 Progen 10523 Mouse 1C7 10523 1:100 Citrate

6. In your Methods section, please give the sources of any cell lines used in your study.

We have added the sources of the cell lines into the methods section. 

In the experiments, we used organoids from five different donors. Furthermore, we performed drug screens in leukemia cell lines. T cell ALL (MOLT16, HSB2, Jurkat) and B cell ALL (Nalm6, REH) cell lines were obtained from DSMZ-German Collection of Microorganisms and Cell Cultures (DSMZ, Braunschweig, Germany).

7. We note that you have a patent relating to material pertinent to this article. Please provide an amended statement of 

Competing Interests to declare this patent (with details including name and number), along with any other relevant declarations relating to employment, consultancy, patents, products in development or modified products etc. Please confirm that this does not alter your adherence to all PLOS ONE policies on sharing data and materials, as detailed online in our guide for authors http://journals.plos.org/plosone/s/competing-interests by including the following statement: "This does not alter our adherence to PLOS ONE policies on sharing data and materials.” If there are restrictions on sharing of data and/or materials, please state these. Please note that we cannot proceed with consideration of your article until this information has been declared.

This information should be included in your cover letter; we will change the online submission 

 form on your behalf.

We have included the statements in our competing interest section of the revised manuscript. We have added the number of the relevant patent: WO2009/022907, WO2010/090513, WO2012/168930. 

8. Please upload a copy of Supporting Information Table S3 which you refer to in your text on page 13.

This was a typo and is Table S2 in the document. We have changed this accordingly. 

Reviewer #1

Major points: 

1. In general, methotrexate toxicity is not constrained to oral mucositis alone. Hepatic and hematological toxicities are also dose-limiting factors. The paper does not address these issues, which are common limitations in dose-intensified therapies. Moreover, leucovorin is a highly soluble drug, and pretreatment may have unintended complications in over-rescuing methotrexate (over-rescue literature should be cited). The paper should be revised to include these limitations, and statement about the use of leucovorin pre-treatment should be modified to reflect the potential down-side of changing rescue strategies.

We agree with the reviewer that hepatotoxicity, nephrotoxicity and hematological toxicities are other dose-limiting factors in high-dose methotrexate treatment. We chose to study oral mucositis as this is one of the most frequently associated toxicities with HD-MTX courses - occurring in ~20% of patients. Since this paper addresses the effect of methotrexate on the oral mucosa (using an oral mucosa epithelial cell model), we did not explore these other toxicities in this current study. Regardless, we did not mean to mitigate the effect of these toxicities, neither claim that the suggested LV pre-treatment would not result in systemic effect that might be negative. To discuss these potential systemic effects, we have added a sentence in both the introduction and the discussion of the revised document to clarify that these other toxicities could be explored further in a similar manner. Organoid cultures of liver- and kidney- cells to perform similar studies have been described in literature. 

Introduction (line 27 – 28): ‘However, patients often suffer from MTX toxicities such as hepatotoxicity, nephrotoxicity, hematological malignancies and mucositis.’ 

Discussion: ‘By making minor adaptations to the culture media that was previously described to culture oral mucosa organoids, we have shown that organoids can be used to study MTX-toxicity in vitro. These modifications can be applied to organoid cultures from other organs, such as kidney or liver organoids. As such, these models can perhaps be used to study relevant MTX-toxicity that is observed in other organs.’ 

Moreover, in the abstract, we have changed the sentence ‘Although effective, the use of MTX often results in oral mucositis, which is characterized by epithelial cell death’ to: ‘Although effective, the use of MTX often results in severe side-effects, including oral mucositis, characterized by epithelial cell death’ (line 12 – line 13) in the revised manuscript. In studying these toxicities, it would be worthwhile to explore the effect of MTX, but also of 6-Mercaptopurine and PEG-Asparaginase, as these chemotherapeutic regimens are often administered within the same treatment course and the different toxicity profiles should be distinguished. 

Secondly, we fully agree with the reviewer that we should be cautious of the possible ‘rescuing’ effects of a leucovorin pretreatment on leukemia cells. We have added more ‘overrescue’ literature into the introduction and have discussed the possible downsides of a leucovorin pretreatment more clearly in the discussion section. We would not like to state that this intervention should be directly applied into clinics, but it is a promising in vitro strategy of which the clinical safety should be assessed first.

Introduction (line 66 - line 69): In contrast, several pediatric ALL studies (38-41) have suggested that Leucovorin rescue therapy decreases toxicity rates, but might be accompanied by an increased risk of relapse in ALL. This phenomenon has been referred to as the folate ‘over-rescue’ principle, where not only healthy cells, but also tumor cells are rescued.

Discussion (line 473 – 474): However, it is important to test clinical safety of the described interventions in order not to decrease anti-leukemic activity of MTX.

Currently, a Leucovorin formula prepared in a concentration of 0.1 mg/mL to apply locally to the oral mucosa is available in our clinics. Before introducing such an application to the clinics, we would need to assess whether locally applied Leucovorin to the oral mucosa could lead to systemic levels interfering with leukemia treatment. Hypothetically, when we would locally administer 5 mL of a formula with 0.1 mg/mL to the oral mucosa of a patient of 10 kg with a distribution volume of 3.2 L/kg, in a scenario where 100% would enter the systemic circulation, this would lead to a concentration of 0.5 mg*3.2*10 kg = 16 ng/mL. Previous studies showed that intravenous administration of 50 mg Leucovorin leads to much higher peak plasma levels around 1 ug/mL with a short plasma half-life of around 30 minutes. A pharmacokinetic study, which assesses the plasma levels in health individuals after application of such an intervention to the oral mucosa could aid in determining clinical safety. 

2. In clinical practice, methotrexate-induced oral mucositis can be somewhat mitigated by the use of oral bicarbonate rinses. Was this condition used as a control variable in the experiments that were performed? If so, what was the relative impact compared to conditions that used leucovorin pre-treatment?

We thank the reviewer for the suggestion of studying the effect of oral bicarbonate rinses on mitigating methotrexate-induced oral mucositis. This intervention was not studied here as previous studies, including several meta-analyses, showed conflicting evidence concerning the effect of oral bicarbonate rinses, even though we know it is often applied in clinics. We agree with the reviewer that this would be a very clean model to study the effect of MTX and oral bicarbonate rinses on the oral mucosa cells. However, for now this was outside the scope of this study. 

3. The Discussion section could be shorter, since some of the same points are re-stated in the Conclusion paragraph.

We agree with the Reviewer that the discussion section of the original manuscript summarized our findings, rather than discussing implication etc. Therefore, we have extensively rewritten the discussion section of our revised manuscript to address this point. We expect this rewriting will resolve the issue raised by the Reviewer. 

Minor points:

1. At line 300, “As, clinically….” appears to be missing a word.

We thank the reviewer for his/her critical reading and have modified this sentence as suggested by the reviewer: “As MTX plasma levels have dropped by 54 hours post infusion in clinics…” to correct this typo. 

2. The authors should define IC50 at its first use.

We have added a definition of IC50 at line 212-213: “IC50 values (=half maximal inhibitory concentration; the concentration at which 50% of cells are dead)…” to correct this issue. 

3. The legend for Figure 4 could be more clearly written to help the reader understand what effects LV rescue have on MTX toxicity in leukemia cell lines.

We agree with the Reviewer that the legends of the Figures could be improved to aid the reader understand the data shown in the Figure. Therefore, we have revised the Legend of this figure in the revised manuscript. In addition, we have also improved the legends for Figure 1 – 4 of the revised manuscript. We hope these modifications will resolve the Reviewer’s concerns.

See for changes: line 618 – 789.

Reviewer #2

1. The drug unit labels are unnecessarily confusing on the viability plots (i.e., Fig. 1E, 2C, etc). It is difficult for most people to figure out what -0.5 log10 uM converts to, and this is the approximate effect size (based in change in IC50) induced by leucovorin treatment. It is fine to show the data on log scale, but it would be much easier for the doses to be shown as 1 uM, 0.1 uM, 0.01 uM, etc.

We agree with this comment of the Reviewer. In the revised manuscript, we have amended all graphs in all the figures of the original manuscript that contain a log-axis according to the Reviewers suggestion. We agree with the Reviewer that this modification makes the graphs easier to interpret for the reader and therefore thank the Reviewer for his/her suggestion. 

2. Why are the number of replicates from different timepoints/conditions so different in many experiments? For example, Fig 1E, 2B, 2E, and several others. This gives the strong impression that the data shown are an amalgamation of different experiments performed at different times. If the experiments were performed simultaneously, this should be clearly stated in the methods. If the data are a mix of experiments at same vs different time points, all of the experiments should be repeated at one time point with the same number of replicates per condition. It is a good idea to repeat experiments to ensure reproducibility, but results of only one representative experiment should be shown.

In the revised manuscript, we have amended the figures to only show matching replicates. With matching, we mean they were obtained in the same experiment (for example IC50 value MTX + 0.05 LV rescue 12 h post MTX matching to the IC50 values MTX +0.05 uM LV rescue 24 hours post MTX since these were values obtained from plates where the MTX exposure was initiated at the same time (in the same 384 well plate) and the LV was added at different timepoints. To further facilitate the reader to understand which IC50 matches to which other (since we show three replicate experiments, we have indicated this with different symbols. Hence, all the IC50 values obtained in one experiment have round symbols, whereas those obtained in a different experiment (performed in a different week for example) are shown with square symbols.

Furthermore, we have removed any control (MTX only) values that could not be matched to other IC50 values (for example obtained after LV rescue) from these graphs. We initially placed these values also in the Figures of the original manuscript, in light of transparency. However, we agree with the Reviewer that these values, since obtained separately from the other datapoints, do not add to the figures and only generate confusion for the reader.

Taken together, we hope these modifications will satisfy the Reviewers comments. 

3. Are the changes in IC50s induced by leucovorin in Fig 1E, 3C, 3E and 3F statistically significant? Significance of all relevant comparisons should be assessed, with appropriate correction for multiple hypothesis testing. The lack of statistical analyses to support the conclusions is a major flaw throughout this paper.

We agree with the reviewer that statistical analyses of the data is of importance and have analyzed the data as described in our new methods section: 

Methods (line 286 - line 293): Raw luminescence values obtained after readout of MTX drug screens, were normalized to the average of untreated controls (n=3, 100%) and staurosporin-treated controls (n=3, 0%), using the formula: (value- 0% control)/(100% control – 0% control)*100%. Resulting percentages of viability were transferred to GraphPad v8 to generate kill curve. Curves were fit using the option ‘log inhibitor vs. normalized response – variable slope. IC50s/AUC values were obtained from GraphPad analysis. The change in IC50 values over time was assessed using a Pearson correlation, performed in Graphpad prism, using XY analysis, correlation option. AUC’s were compared as previously described.

We have added the correlations and p-values in our results and figures. The addition of these statistics does not alter our conclusions. 

4. LV pre-treatment also protects leukemia cells. The authors claim that this is less than in normal oral mucosa, but again there is no statistical evidence to support that this is a significant difference. FURTHERMORE, IT IS GLARING THAT THE PRE-TREATMENT OF LEUKEMIA CELLS IS DONE WITH A DOSE OF LEUKOCORIN 10-FOLD LOWER THAN THAT OF THE ORGANOIDS!!! As indicated on the plots of Fig 3B, PT = 0.1 uM in organoids, and Fig 4, 0.01 uM leukemia cells. Is it any surprise that there is less rescue with a lower dose of leucovorin?? This is viewed as a completely unacceptable attempt to mislead the reviewers and readers, and in the opinion of this reviewer this alone is grounds for rejection of the paper. Unless perhaps this is a typo, but then this raises concerns about rigor elsewhere in this manuscript.

We understand the concern of the reviewer, and would like to sincerely apologize for this typo, which we have amended in the revised manuscript Of course, we would not have made these statements if the used concentration of LV was indeed 10 times lower, since we have written this manuscript to share our findings with the research community, and thus would not at all attempt to mislead the reader in any way. Although we understand the reviewers concern, we have gone over the remainder of the manuscript to assure no other typos are present, and can assure him/her that we would not purposely introduce such differences between the controls of our experiments.

Reviewer #3 

1. First of all, the authors should clarify the state of the mucosal cells during the experiment. Are they isolated as single cells, which seems to be the case since the authors report they trypsinize them just prior to adding drugs? Or do they have time to assemble into the 3d organoids, as mentioned in the title? 

This study was performed with organoids derived from human oral mucosa resections. These cultures are established from tissue digest. Once established, cultures can be passaged regularly, by digestion (using TrypLE) of the formed 3D organoids into smaller fragments (single cells/clumps of cells). Once replated, these single cells/clumps of cells, will proliferate and form new organoids, which contain both proliferating basal cells, and differentiated keratinocytes (see Figure 1A). In this study, MTX toxicity was determined on established organoid cultures (>3 passages), not fresh tissue isolates. 

In the method section of the paper, we state that organoids were split to single cells 2 days prior to the start of drug screening. This means that by the time of screening, organoids have already reformed from these single cells, to form 3D structures. In the revised manuscript, we now state this more clearly: ‘Two days later, when single cells already reformed into small organoids, the organoids were collected from the BME by addition of 1 mg/mL dispase II (Sigma-Aldrich, cat.no. D4693) to the medium of the organoids.’ 

Regardless, since this was apparently not made clear by us in the original manuscript, we have modified the introduction section on organoids in our revised manuscript. It now contains the following section: 

Introduction (line 88 - line 91): ‘Recently, we described an organoid model derived from healthy oral mucosa (34). The resulting patient-derived structures consist of a functional stratified squamous epithelium that can be maintained and expanded in culture for over six months. *NEW SECTION STARTING* Upon passaging, organoids grown from primary oral mucosa tissue, can be broken into smaller fragments, which will proliferate and result in the formation of new organoids. As such, organoid technology allows us to multiply human wildtype epithelial cells for a wide variety of applications, including drug screening.’ 

Taken together, we hope these modifications will satisfy the Reviewers concern.

2. Secondly, there is little or no discussion of why this system is superior to testing of individual isolated cell cultures, and so it would be good to comment on that. Do the authors think that penetration of drugs into the interior of 3D organoids is an important factor?

In the introduction and in the answers above, we have more clearly described the advantages of a 3D model as opposed to single cell cultures. 

The penetration of Methotrexate into intracellular compartments is mediated through different transporters, of which the RFC1, the PCFT and the FR receptors are most important. As methotrexate is administered intravenously in pediatric ALL patients, the main mode of transportation is through the RFC1 transporter, which is both located apically as basolaterally, whereas for instance the PCFT transporter is mainly located apically in the gastro-intestinal tract. As transporters are located at all sides of the organoid, we expect that penetration of the drugs will not hamper results in a 3D-organoid model. 

We hope that we have addressed the concerns of the reviewers sufficiently and are looking forward to hear your decision concerning the revised manuscript. 

Kind regards, 

Prof.Dr. Marry M. van den Heuvel-Eibrink

---

## [Decision Letter · Decision Letter 1]

27 Mar 2020

Patient-derived oral mucosa organoids as an in vitro model for methotrexate induced toxicity in pediatric acute lymphoblastic leukemia

PONE-D-19-32380R1

Dear Dr. Oosterom,

We are pleased to inform you that your manuscript has been judged scientifically suitable for publication and will be formally accepted for publication once it complies with all outstanding technical requirements.

With kind regards,

Obul Reddy Bandapalli, MSc, PhD

Academic Editor

PLOS ONE

Additional Editor Comments (optional):

Reviewers' comments:

Reviewer's Responses to Questions

**Comments to the Author**

1. If the authors have adequately addressed your comments raised in a previous round of review and you feel that this manuscript is now acceptable for publication, you may indicate that here to bypass the “Comments to the Author” section, enter your conflict of interest statement in the “Confidential to Editor” section, and submit your "Accept" recommendation.

Reviewer #1: All comments have been addressed

Reviewer #2: All comments have been addressed

Reviewer #3: All comments have been addressed

2. Is the manuscript technically sound, and do the data support the conclusions?

Reviewer #1: Yes

Reviewer #2: Yes

Reviewer #3: Yes

3. Has the statistical analysis been performed appropriately and rigorously? 

Reviewer #1: Yes

Reviewer #2: Yes

Reviewer #3: Yes

4. Have the authors made all data underlying the findings in their manuscript fully available?

Reviewer #1: Yes

Reviewer #2: Yes

Reviewer #3: Yes

5. Is the manuscript presented in an intelligible fashion and written in standard English?

Reviewer #1: Yes

Reviewer #2: Yes

Reviewer #3: Yes

6. Review Comments to the Author

Reviewer #1: (No Response)

Reviewer #2: The authors have addressed all of my concerns. My major critique turned out to be a typo that has been corrected.

Reviewer #3: none

7. PLOS authors have the option to publish the peer review history of their article (what does this mean?). If published, this will include your full peer review and any attached files.

Reviewer #1: No

Reviewer #2: No

Reviewer #3: No

---

## [Editor Report · Acceptance letter]

4 May 2020

PONE-D-19-32380R1 

Patient-derived oral mucosa organoids as an *in vitro* model for methotrexate induced toxicity in pediatric acute lymphoblastic leukemia 

Dear Dr. Oosterom:

I am pleased to inform you that your manuscript has been deemed suitable for publication in PLOS ONE. Congratulations! Your manuscript is now with our production department. 

With kind regards,

on behalf of

Dr. Obul Reddy Bandapalli 

Academic Editor

PLOS ONE